# Diagnosis and Treatment of Bronchopulmonary Lophomoniasis in a Patient with Persistent Granuloma: A Case Report

**DOI:** 10.3390/reports7040102

**Published:** 2024-11-19

**Authors:** Antonio Mier-Briseño, Eloísa Ramírez-Alanís, Miguel Armando Benavides-Huerto, Francisco Alejandro Lagunas-Rangel

**Affiliations:** 1Department of Pneumology, Morelia Medical Clinic, Morelia 58260, Michoacán, Mexico; 2Clinical Analisys Laboratory, Sanatorio La Luz, Morelia 58260, Michoacán, Mexico; 3Laboratory of Pathology and Cytopathology “Dr. Miguel Benavides”, Morelia 58260, Michoacán, Mexico; 4Department of Genetics and Molecular Biology, Centro de Investigación y de Estudios Avanzados del Instituto Politécnico Nacional, Mexico City 07360, Mexico; 5Department of Surgical Sciences, Uppsala University, 752 36 Uppsala, Sweden

**Keywords:** parasite, trophozoite, hemoptysis, granuloma, metronidazole

## Abstract

**Background and Clinical Significance**: *Lophomonas* is a multiflagellate anaerobic protozoan that usually inhabits the intestines of insects, mainly cockroaches. However, bronchopulmonary infections caused by this parasite have been increasingly reported worldwide in recent decades. We provide important information for the diagnosis of this disease, which often goes undetected and frequently leads to misdiagnosis and inadequate treatment. It is noteworthy that this is the third case reported in Mexico. **Case Presentation**: A 37-year-old male patient was hospitalized several times for pneumonia with a persistent granuloma in the right bronchial lobe. After extensive testing, the patient was diagnosed with bronchopulmonary lophomoniasis and successfully treated with metronidazole. **Conclusions**: Clinicians worldwide should be aware of the existence of lophomoniasis, especially in low-income regions with poor sanitation and high insect exposure. This parasitic infection, although rare, may be underestimated due to its nonspecific respiratory symptoms, which may mimic other infections.

## 1. Introduction and Clinical Significance

*Lophomonas* spp. is a multiflagellate anaerobic parasitic protozoan belonging to the phylum Parabasalia. It resides in the intestines of certain insects, mainly cockroaches, as part of their microbiome. The life cycle of *Lophomonas* spp. is not yet fully understood. Currently, only two stages of this parasite are known: the trophozoite stage, which is the active feeding form within the host, and the cyst form [1].

In recent decades, an increase in bronchopulmonary infections caused by this parasite has been reported worldwide [2]. Humans can become infected by inhaling cysts excreted in the fecal matter of these insects. Upon inhalation of the cyst, a multiflagellate trophozoite emerges that firmly adheres to the respiratory mucosa and secretes proteases that can cause chronic asthmatic-type inflammation [3]. The pathogenic mechanisms of *Lophomonas spp*. are still poorly understood [4], but it is likely that the parasite alters the structure and function of respiratory epithelial cells, causing inflammation. This inflammation, in turn, may contribute to respiratory symptoms and tissue damage, playing a key role in the disease process.

Clinical Significance: The present report provides information that could be useful in the diagnosis of lophomoniasis, a parasitic disease frequently overlooked by both clinicians and routine laboratory tests due to its rarity and nonspecific clinical presentation. This case is particularly noteworthy as it presents the third documented case in Mexico, underscoring the importance of raising awareness and improving diagnostic protocols.

## 2. Case Presentation

A 37-year-old married office worker was hospitalized for pneumonia. He reported weight loss and a nine-month history of morning cough with mucohemoptotic and sometimes mucopurulent sputum, but no fever. The patient did not appear to be immunosuppressed, as evidenced by a normal white blood cell count of 6840 leukocytes/μL, with a differential of 73% segmented neutrophils, 24% lymphocytes, 2% monocytes, 0.8% eosinophils, and 0.2% basophils. Additionally, an HIV test was negative. Chest scan revealed a pulmonary nodule in the right upper lobe (RUL) measuring 3.7 × 3.3 cm with a thin smooth wall and central oval cavitation (Figure 1).

Adjacent to it, in the posterior segment of the RUL, there was an area of consolidation with cystic bronchiectasis measuring 7.5 × 5.3 cm and extending from the periphery to the ipsilateral region, dilating the right upper bronchus. The rest of the lung parenchyma appeared normal. Tuberculosis tests (acid-fast bacilli [AFB] staining and cultures) were negative. He was treated with clindamycin, levofloxacin, benzonatate, and omeprazole. He improved and was discharged after two days with a follow-up appointment in one week, which he did not attend.

Two months later, the patient returned to the clinic with a sore throat caused by continuous episodes of cough with hemoptysis during the previous eight days. He also presented with fever, pain in both scapular regions, wheezing, substernal heartburn, and exertional dyspnea. The patient had leukocytosis (15,170 leukocytes/μL). Differential count showed 83% segmented neutrophils, 14% lymphocytes, 2% monocytes, 0.7% eosinophils, and 0.3% basophils. A second HIV test was negative. Chest scan again showed nodulation in the RUL with an air-fluid level causing an adjacent area of pneumonitis. The hilum was displaced to the right and upward, without cardiomegaly. He was hospitalized for hemoptysis and underwent bronchoscopy, bronchial lavage, aspiration, and biopsy. The pathology report indicated chronic granulomatous pneumonia in the RUL, compatible with a tuberculous etiology, without malignant cells (Figure 2A,B). However, tuberculosis tests (AFB staining and cultures) again were negative. Bronchial lavage and bronchial brushing revealed intense inflammatory changes. The patient received treatment with itraconazole, levofloxacin, ipratropium bromide, and salbutamol, and was discharged one week later. In a follow-up within one week of discharge, the patient was in good clinical condition, with decreased coughing spells. He sometimes presented with mucoid sputum, occasionally sprinkled with a small amount of blood, but reported significant clinical improvement.

After one month, the patient returned with new attacks of cough and frank hemoptysis, but without fever, chest pain, or dyspnea on exertion. A chest scan showed the persistent cavitation and infiltration in the posterior segment of the RUL, radiating to the hilum, with the mediastinum slightly shifted to the right and upward. The rest of the examination was normal. The previous treatment was continued, adding etamsylate. After two weeks, there was no hemoptysis or fever, but the patient experienced flu-like symptoms with respiratory distress. He presented with pink mucopurulent sputum and pain in the right anterior hemithorax. Tuberculosis test (AFB staining, QuantiFERON [QFT] and cultures) were negative. At this point, the patient was treated with rifampicin, isoniazid, levofloxacin, ipratropium bromide, salbutamol, and dextromethorphan. At the follow-up visit, the patient reported a recurrence of cough with hemoptysis, but no fever or dyspnea. A chest scan showed a decreased perilesional infiltrate in the RUL, with less pulmonary condensation and less cavitation. There were no new consolidations or pleural effusions. The right hilum was slightly retracted upward and to the right, without cardiomegaly. Treatment was maintained with rifampicin, isoniazid, ipratropium bromide, and salbutamol, and voriconazole was added.

After two weeks, the patient presented fever that was relieved with paracetamol. He had no hemoptysis, but continued pain in the right anterior and posterior hemithorax, cough with yellowish sputum, significant dyspnea, and apparent weight loss of 4 kg. The patient had leukocytosis (13,320 leukocytes/μL). Differential count showed 79% segmented neutrophils, 19% lymphocytes, 1% monocytes, 0.7% eosinophils, and 0.3% basophils. A chest scan showed recurrent pulmonary consolidation and cavitations in the posterior segment of the RUL, extending from the hilum. In contrast, the left lung was greatly expanded. Videobronchoscopy was performed and, using the highest magnification (100X), evidence of motile microorganisms was observed. Subsequent microscopic examination of fresh bronchial lavage revealed *Lophomonas* trophozoites (Figure 3A and Appendix A). These findings were confirmed by pathological studies (Figure 3B,C), which finally led to the diagnosis of bronchopulmonary lophomoniasis. The *Lophomonas* small ribosomal subunit (SSU) rRNA PCR test performed on the bronchial lavage precipitate sample also yielded a positive result (4). The patient was treated with metronidazole. After one month of follow-up, pulmonary consolidation showed marked improvement and reduction (Figure 3D). The patient is in good condition and improving.

## 3. Discussion

Cases of lophomoniasis have been reported in 10 countries on four continents (Asia, America, Europe, and Africa), with the majority of cases coming from Asia. Iran accounts for the largest number of reported cases (77.2%), followed by China (9.3%), Panama (6.2%), Turkey (3.6%), India (1.2%), Mexico (0.9%, including this case), Spain (0.6%), Egypt (0.3%), Malaysia (0.3%), and Peru (0.3%) [2]. Remarkably, this is the third case reported in Mexico. The global prevalence of lophomoniasis is believed to be higher than reported, as many cases are not diagnosed [1]. Lophomoniasis mainly affects young adults like our patient and does not show significant differences between sexes.

The differential diagnosis of lophomoniasis is particularly difficult due to the limited data available on this infection and the absence of a practical diagnostic guide. This paucity of information makes it difficult to distinguish lophomoniasis from other respiratory infections with similar presentations. However, from our point of view, this should include conditions such as tuberculosis, bacterial pneumonia (e.g., *Streptococcus pneumoniae*, *Staphylococcus aureus*, *Haemophilus influenzae*), fungal pneumonia (e.g., *Aspergillus*), *Pneumocystis jirovecii* infection, asthma, and chronic bronchitis, among other respiratory diseases presenting with similar symptoms. It is important to be aware of lophomoniasis, especially in cases of refractory pulmonary infections, even in non-immunosuppressed patients. This is especially relevant for individuals at higher risk of exposure, such as those living in unsanitary conditions or in close proximity to insect vectors such as cockroaches. It has been mentioned that the prevalence of *Lophomonas* infection was relatively high among suspected tuberculosis and bacterial pneumonia patients, probably due to the similarity of the clinical symptoms [5,6,7]. Furthermore, co-infections of *Lophomonas* with other pathogens, including bacteria, viruses, and parasites such as *Mycobacterium* [8], SARS-CoV-2 [9], *Pneumocystis jirovecii* [10], and *Echinococcus* [11] have been documented. These co-infections can significantly complicate both the diagnostic process and clinical management due to overlapping symptoms.

The *Lophomonas* parasite can infect both the upper respiratory tract (mainly the paranasal sinuses) and the lower respiratory tract (mainly the lungs and bronchi). The most frequent presenting symptoms are cough (70.69%), fever (60.35%), expectoration (46.55%), dyspnea (41.38%), and chest pain (29.31%), among others [1-3,12]. Our patient, in addition to these symptoms, presented hemoptysis. Common comorbidities include various types of cancer, allergies, autoimmune diseases, immunodeficiencies, chronic obstructive pulmonary disease (COPD), chronic renal failure, diabetes mellitus, hypertension, pulmonary fibrosis, and bacterial, viral, and parasitic infections such as aspergilloma, *Candida* species, SARS-CoV-2, cryptococcosis, cytomegalovirus, echinococcosis, *Klebsiella* species, *Pneumocystis jirovecii*, *Pseudomonas aeruginosa*, and tuberculosis [1]. Bronchoalveolar lavage samples are the most commonly used to detect *Lophomonas* compared to other respiratory samples. The most prevalent diagnostic method is microscopic examination, used in 100% of reported cases, with some cases confirmed by molecular methods such as PCR (34.48%) [1,2]. In this case, the diagnosis was initially made through microscopy of bronchial lavage and biopsy samples, then confirmed by PCR. The *Lophomonas* trophozoite is round-ovoid or pyriform, measuring 20–60 µm long and 12–20 µm wide. The cytoplasm is granular and contains some phagocytized food particles. In the apical zone, the protozoan has a plume of numerous flagella with an irregular orientation. The outer flagella are smaller, separate, and vibrate freely in the surrounding fluid medium. Occasionally, the nucleus can be observed as a round, dark body located just below the insertion of the flagella [3,12].

Metronidazole is the drug of choice to treat lophomoniasis, with satisfactory results in most cases [1,2], as in the one presented here. Another reported treatment option includes tinidazole, which has shown good results in some cases [13]. In addition, combinations of metronidazole with other antiparasitics, such as ornidazole [1] and albendazole [14], have been used with variable results. These combination therapies are mainly considered for patients coinfected with other microorganisms or who do not respond to metronidazole alone. In terms of dosing, metronidazole is usually administered at 500–750 mg every 8 h for 7 to 14 days [15,16], while tinidazole is usually administered at 500 mg every 12 h for a similar period [13]. However, there is considerable variation in dosing regimens between studies and clinical cases, reflecting the lack of standardized guidelines. It is recommended that future studies focus on optimizing dosing and duration of treatment to improve therapeutic outcomes and establish more consistent protocols.

## 4. Conclusions

Physicians worldwide should be aware of the diagnosis of lophomoniasis, especially in developing and low-income countries, where the risks of exposure are higher due to unsanitary conditions and close contact with insect vectors such as cockroaches. This parasitic infection, although rare, may be underestimated due to its nonspecific respiratory symptoms, which may mimic other infections.

## Figures and Tables

**Figure 1 reports-07-00102-f001:**
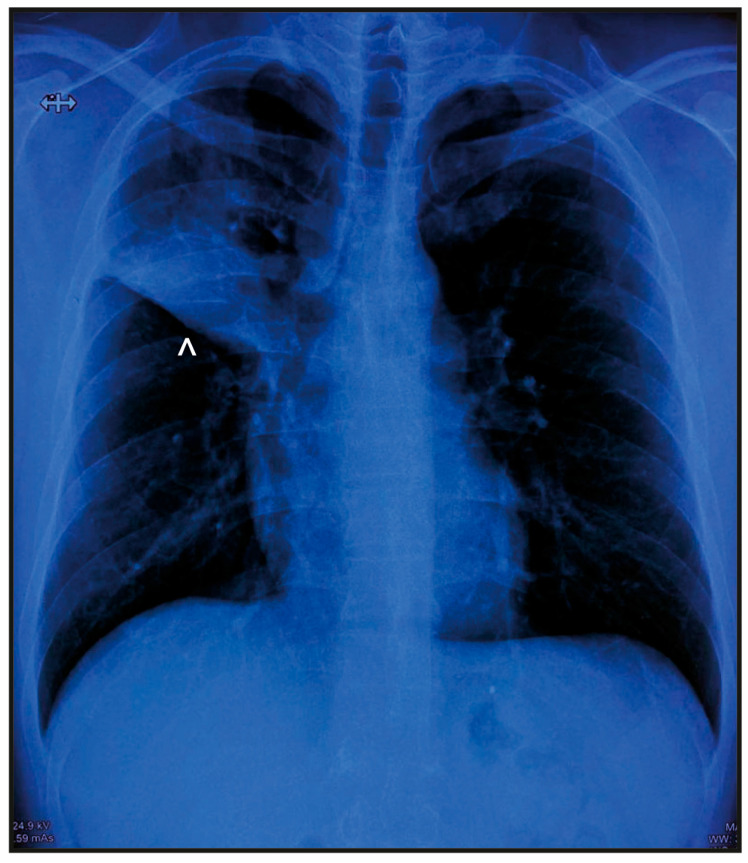
Patient characteristics at the time of case presentation. Chest scan showing a nodule in the right upper lobe with an area of consolidation accompanied by cystic bronchiectasis (indicated by arrowhead).

**Figure 2 reports-07-00102-f002:**
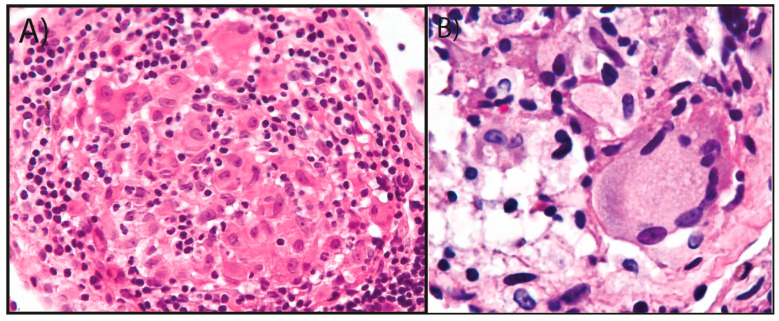
First histopathological findings. (**A**) Histopathological examination showed a granuloma without necrosis or fungus (H&E, 20X). (**B**) Microscopic analysis revealed an idiopathic granuloma (H&E, 40X).

**Figure 3 reports-07-00102-f003:**
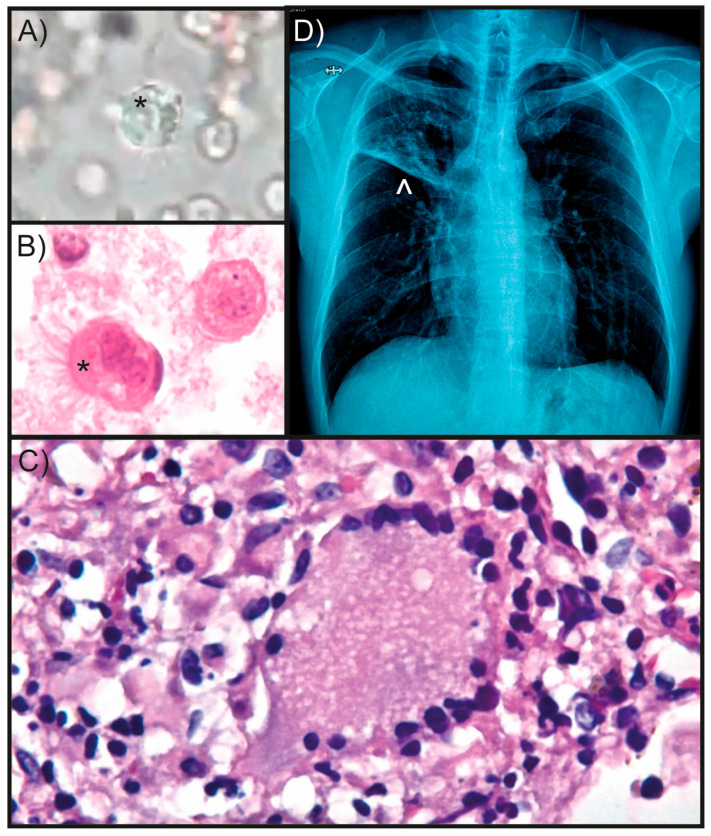
Patient characteristics at the time of diagnosis and after treatment. (**A**) *Lophomonas* trophozoite identified in bronchial lavage fluid (asterisk). (**B**) Cytological study confirmed the presence of *Lophomonas* spp. trophozoites (asterisk) (H&E, 100X). Bronchial lavage fluid was collected, fixed with 70% ethyl alcohol, and then centrifuged to concentrate the sample. The resulting precipitate was placed on a slide, stained, and subsequently examined under the microscope. (**C**) Granuloma without necrosis or fungus (H&E, 40X). (**D**) Chest scan performed one month after metronidazole treatment showing a decrease in nodule size and area of consolidation in the right upper lobe.

## Data Availability

The original data presented in the study are included in the article, further inquiries can be directed to the corresponding author.

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
