# Peer review of "Diagnosis and Treatment of Bronchopulmonary Lophomoniasis in a Patient with Persistent Granuloma: A Case Report"

_reports, 2024, doi:10.3390/reports7040102_

Round 1

Reviewer 1 Report

Comments and Suggestions for Authors

1.        This paper should aim to enhance the steps and theoretical basis of differential diagnosis, particularly addressing initial misdiagnoses, offering valuable insights from the authors' academic and professional expertise.

2.       The authors could discuss how Lophomonas infections can be distinguished from respiratory diseases caused by other common pathogens and specify how clinicians can differentiate these conditions during initial examinations.

3.        The authors could include data on the pathogen’s biological characteristics. Supplementing background information on Lophomonas spp., such as its life cycle, modes of transmission, pathogenic mechanisms, and reasons for treatment challenges, would be beneficial.

4.        The authors should highlight the unique aspects of this case, the third reported in Mexico, including unusual symptoms or challenging treatment paths, to underscore the distinctive nature of this specific case.

5.       While metronidazole proved effective in this instance, it would be advantageous to explore potential alternative treatment strategies or combinations for patients who may exhibit resistance to or intolerance of this medication. Additionally, offering recommendations grounded in current literature regarding dosage optimization and treatment duration would be valuable.

Author Response

Dear Reviewer, 

We express our sincere gratitude for your constructive comments and for the time you have taken to review our manuscript. We have worked diligently to improve the quality of the manuscript in accordance with your valuable comments. Below you will find my responses to your comments, highlighted in blue text.

Comment 1 and 2.

  1. This paper should aim to enhance the steps and theoretical basis of differential diagnosis, particularly addressing initial misdiagnoses, offering valuable insights from the authors' academic and professional expertise.
  2. The authors could discuss how Lophomonas infections can be distinguished from respiratory diseases caused by other common pathogens and specify how clinicians can differentiate these conditions during initial examinations.

Response 1 and 2.

We add a paragraph in the discussion that refers to what you mention and appears in the manuscript as follows:

The differential diagnosis of lophomoniasis is particularly difficult due to the limited data available on this infection and the absence of a practical diagnostic guide. This paucity of information makes it difficult to distinguish lophomoniasis from other respiratory infections with similar presentations. However, from our point of view, this should include conditions such as tuberculosis, bacterial pneumonia (e.g., Streptococcus pneumoniae, Staphylococcus aureus, Haemophilus influenzae), fungal pneumonia (e.g., Aspergillus), Pneumocystis jirovecii infection, asthma, and chronic bronchitis, among other respiratory diseases presenting with similar symptoms. It is important to be aware of lophomoniasis, especially in cases of refractory pulmonary infections, even in non-immunosuppressed patients. This is especially relevant for individuals at higher risk of exposure, such as those living in unsanitary conditions or in close proximity to insect vectors such as cockroaches. It has been mentioned that the prevalence of Lophomonas infection was relatively high among suspected tuberculosis and bacterial pneumonia patients, probably due to the similarity of the clinical symptoms [6–8]. Furthermore, co-infections of Lophomonas with other pathogens, including bacteria, viruses and parasites such as Mycobacterium [9], SARS-CoV-2 [10], Pneumocystis jirovecii [11] and Echinococcus [12] have been documented. These co-infections can significantly complicate both the diagnostic process and clinical management due to overlapping symptoms.

Comment 3.

  1. The authors could include data on the pathogen’s biological characteristics. Supplementing background information on Lophomonas spp., such as its life cycle, modes of transmission, pathogenic mechanisms, and reasons for treatment challenges, would be beneficial.

Response 3.

The aspects you mention were added to the introduction, which appears as follows in the manuscript:

Lophomonas spp. is a multiflagellate anaerobic parasitic protozoan belonging to the phylum Parabasalia. It resides in the intestines of certain insects, mainly cockroaches, as part of their microbiome [1]. The life cycle of Lophomonas spp. is not yet fully understood. Currently, only two stages of this parasite are known: the trophozoite stage, which is the active feeding form within the host, and the cyst form [2].

In recent decades, an increase in bronchopulmonary infections caused by this parasite has been reported worldwide [3]. Humans can become infected by inhaling cysts excreted in the fecal matter of these insects. Upon inhalation of the cyst, a multi-flagellate trophozoite emerges that firmly adheres to the respiratory mucosa and se-cretes proteases that can cause chronic asthmatic-type inflammation [4]. The pathogenic mechanisms of Lophomonas spp. are still poorly understood [5], but it is likely that the parasite alters the structure and function of respiratory epithelial cells, causing in-flammation. This inflammation, in turn, may contribute to respiratory symptoms and tissue damage, playing a key role in the disease process.

Comment 4.

  1. The authors should highlight the unique aspects of this case, the third reported in Mexico, including unusual symptoms or challenging treatment paths, to underscore the distinctive nature of this specific case.

Response 4.

Throughout the manuscript we have tried to highlight the unique aspects of this case.

Comment 5.

  1. While metronidazole proved effective in this instance, it would be advantageous to explore potential alternative treatment strategies or combinations for patients who may exhibit resistance to or intolerance of this medication. Additionally, offering recommendations grounded in current literature regarding dosage optimization and treatment duration would be valuable.

Response 5.

Thank you for your valuable input. Everything you have mentioned has been incorporated into the manuscript and appears as follows at the end of the discussion:

Metronidazole is the drug of choice to treat lophomoniasis, with satisfactory results in most cases [1,2], as in the one presented here. Other reported treatment option in-cludes tinidazole, which has shown good results in some cases [5]. In addition, combi-nations of metronidazole with other antiparasitics, such as albendazole [6], have been used with variable results. These combination therapies are mainly considered for pa-tients coinfected with other microorganisms or who do not respond to metronidazole alone. In terms of dosing, metronidazole is usually administered at 500-750 mg every 8 hours for 7 to 14 days [7,8], while tinidazole is usually administered at 500 mg every 12 hours for a similar period [5]. However, there is considerable variation in dosing regi-mens between studies and clinical cases, reflecting the lack of standardized guidelines. It is recommended that future studies focus on optimizing dosing and duration of treatment to improve therapeutic outcomes and establish more consistent protocols.

Reviewer 2 Report

Comments and Suggestions for Authors

It is a case repot about a rare and so far, mostly unknown disease. The abstract is concise. The report itself is well structured. The references are not abundant but focus on the latest publications to the topic. The case description provides all the necessary information about the course of the disease and is supported by informative images. The discussion resumes the basic and important facts needed by clinicians. Since this is a case report and not a review, I think the scope of the article, the number of references and the detail of the description of the disease is completely sufficient. Of course, there are already case reports on lophomoniasis, but due to the rarity of the disease and the limited data available, I consider every case report to be a valuable contribution, that also raises awareness of the existence of this disease.

Author Response

Dear Reviewer, 

We express our sincere gratitude for your constructive comments and for the time you have taken to review our manuscript. We have worked diligently to improve the quality of the manuscript in accordance with your valuable comments. Below you will find my responses to your comments, highlighted in blue text.

Comment 1.

It is a case repot about a rare and so far, mostly unknown disease. The abstract is concise. The report itself is well structured. The references are not abundant but focus on the latest publications to the topic. The case description provides all the necessary information about the course of the disease and is supported by informative images. The discussion resumes the basic and important facts needed by clinicians. Since this is a case report and not a review, I think the scope of the article, the number of references and the detail of the description of the disease is completely sufficient. Of course, there are already case reports on lophomoniasis, but due to the rarity of the disease and the limited data available, I consider every case report to be a valuable contribution, that also raises awareness of the existence of this disease.

Response 1.

It is nice to know that you notice the relevance of our manuscript and we appreciate your support in publishing it.

Reviewer 3 Report

Comments and Suggestions for Authors

Comment

This paper describes a rare case of bronchopulmonary lophomoniasis, which was successfully treated. As medical treatment advances and the number of immunosuppressed patients increases, it is necessary to keep lophomoniasis in mind when treating refractory lung infections. This report is a valuable tool in raising awareness of this condition.

Medical examinations were performed following repeated complaints, and a challenging diagnosis was made. There are some lessons to be learned from the clinical findings. In particular, microscopic observation of the sample enabled differentiation from bronchial ciliated cells. For these reasons, we feel this paper should be published.

However, the path to the diagnosis was complex, and the patient appeared to respond to various treatments several times. The question is whether this symptoms are monistic ones. If you have other microbial test results, they should be submitted as negative data.

Currently, infections have been reported in healthy people with no immune disorders, so it is advisable to add information about the patient’s background.

Minor comments

    The chest X-ray in Fig. 1 was taken at the time of the initial consultation, and the pathological findings are from a specimen obtained through bronchoscopy 2 months later. These findings are taken in different time phases, so it would be better to separate them.

    Line 57: What is a phymic granuloma? Is phymic Spanish?

    Line 65: If it is of tuberculous etiology, it may be accompanied by caseous necrosis.

    Line 88: Specific product names (e.g., Dotbal S and Combivent) should be avoided. Please change to generic names.

    Line 100: How was the image taken using videobronchoscopy? Did you just look at the BAL fluid?

    Line 102: The photograph in Fig. 2B shows what specimens were stained, how they were treated, and how they were examined under a microscope.

    Line 130: You say "microscopy of bronchial lavage." Does this refer to the videobronchoscopy in Line 100?

Author Response

Dear Reviewer, 

We express our sincere gratitude for your constructive comments and for the time you have taken to review our manuscript. We have worked diligently to improve the quality of the manuscript in accordance with your valuable comments. Below you will find my responses to your comments, highlighted in blue text.

Comment 1.

This paper describes a rare case of bronchopulmonary lophomoniasis, which was successfully treated. As medical treatment advances and the number of immunosuppressed patients increases, it is necessary to keep lophomoniasis in mind when treating refractory lung infections. This report is a valuable tool in raising awareness of this condition.

Medical examinations were performed following repeated complaints, and a challenging diagnosis was made. There are some lessons to be learned from the clinical findings. In particular, microscopic observation of the sample enabled differentiation from bronchial ciliated cells. For these reasons, we feel this paper should be published.

However, the path to the diagnosis was complex, and the patient appeared to respond to various treatments several times. The question is whether this symptoms are monistic ones. If you have other microbial test results, they should be submitted as negative data.

Response 1. 

As stated in the manuscript, various tests were conducted to detect tuberculosis, including acid-fast bacillus (AFB) staining, QuantiFERON assays, and cultures. In addition, conventional cultures were performed to identify other potential pathogens.

Comment 2.

Currently, infections have been reported in healthy people with no immune disorders, so it is advisable to add information about the patient’s background.

Response 2.

As you suggest we added more information about the patient's background at the beginning of the clinical case and it appears as follows in the manuscript:

A 37-year-old married office worker was hospitalized for pneumonia. He reported weight loss and a nine-month history of morning cough with mucohemoptotic and sometimes mucopurulent sputum, but no fever. The patient did not appear to be im-munosuppressed, as evidenced by a normal white blood cell count of 6,840 leuko-cytes/μL, with a differential of 73% segmented neutrophils, 24% lymphocytes, 2% monocytes, 0.8% eosinophils, and 0.2% basophils. Additionally, an HIV test was nega-tive. Chest scan revealed a pulmonary nodule in the right upper lobe (RUL) measuring 3.7x3.3 cm with a thin smooth wall and central oval cavitation (Figure 1).

Comment 3.

Minor comments

①    The chest X-ray in Fig. 1 was taken at the time of the initial consultation, and the pathological findings are from a specimen obtained through bronchoscopy 2 months later. These findings are taken in different time phases, so it would be better to separate them.

Response 3.

As per your suggestion, the figures have been separated and are now presented as Figure 1 and Figure 2.

Comment 4.

②    Line 57: What is a phymic granuloma? Is phymic Spanish?

Response 4.

It was corrected

Comment 5.

③    Line 65: If it is of tuberculous etiology, it may be accompanied by caseous necrosis.

Response 5.

Yes, this is correct, but there are Mycobacterium species that do not cause caseous necrosis, and there is also evidence that patients with suspected tuberculosis have lophomonas or co-infection with Lophomonas. This is why several additional tests for tuberculosis were done.

Comment 6. 

④    Line 88: Specific product names (e.g., Dotbal S and Combivent) should be avoided. Please change to generic names.

Response 6.

It was changed as you suggested.

Comment 7.

⑤    Line 100: How was the image taken using videobronchoscopy? Did you just look at the BAL fluid?

Response 7.

We made an error in our initial description. During videobronchoscopy, we observed evidence of motile microorganisms using the maximum magnification (100x). However, the definitive diagnosis was established after microscopic analysis of the fresh bronchial lavage specimen. This was corrected in the manuscript which appears as follows:

Videobronchoscopy was performed and, using the highest magnification (100x), evi-dence of motile microorganisms was observed. Subsequent microscopic examination of fresh bronchial lavage revealed Lophomonas trophozoites (Figure 2A and Supplementary video 1). These findings were confirmed by pathological studies (Figures 2B and 2C), which finally led to the diagnosis of bronchopulmonary lophomoniasis.

Comment 8.

⑥    Line 102: The photograph in Fig. 2B shows what specimens were stained, how they were treated, and how they were examined under a microscope.

Response 8.

It is now Figure 3 and the information you request has been added to its legend.

Figure 3. Patient characteristics at the time of diagnosis and after treatment. A) Lophomonas trophozoite identified in videobronchoscopy (asterisk). B) Cytological study confirmed the presence of Lophomonas spp. trophozoites (asterisk) (H&E, 100X). Bronchial lavage fluid was col-lected, fixed with 70% ethyl alcohol, and then centrifuged to concentrate the sample. The resulting precipitate was placed on a slide, stained, and subsequently examined under the microscope. C) Granuloma without necrosis or fungus (H&E, 40X). D) Chest scan performed one month after metronidazole treatment showing a decrease in nodule size and area of consolidation in the right upper lobe.

Comment 9.

⑦    Line 130: You say "microscopy of bronchial lavage." Does this refer to the videobronchoscopy in Line 100?

Response 9.

No, it refers to microscopic examination of fresh bronchial lavage

Round 2

Reviewer 1 Report

Comments and Suggestions for Authors

No more question